# Purinergic Signaling in Non-Parenchymal Liver Cells

**DOI:** 10.3390/ijms25179447

**Published:** 2024-08-30

**Authors:** Esperanza Mata-Martínez, María Guadalupe Ramírez-Ledesma, Genaro Vázquez-Victorio, Rolando Hernández-Muñoz, Mauricio Díaz-Muñoz, Francisco G. Vázquez-Cuevas

**Affiliations:** 1Departamento de Biología Celular y Desarrollo, Instituto de Fisiología Celular, Universidad Nacional Autónoma de México (UNAM), Ciudad Universitaria, Mexico City 04510, Mexico; espemmtz@gmail.com (E.M.-M.); rhernand@ifc.unam.mx (R.H.-M.); 2Departamento de Neurobiología Celular y Molecular, Instituto de Neurobiología, Universidad Nacional Autónoma de México (UNAM), Boulevard Juriquilla #3001, Querétaro 76230, Mexico; paloma.ledesma@hotmail.com (M.G.R.-L.); mdiaz@comunidad.unam.mx (M.D.-M.); 3Departamento de Física, Facultad de Ciencias, Universidad Nacional Autónoma de México (UNAM), Circuito Exterior S/N, Ciudad Universitaria, Mexico City 04510, Mexico; genvazquez@ciencias.unam.mx

**Keywords:** hepatic stellate cells, Kupffer cells, liver sinusoidal endothelial cells, purinergic signaling, purinergic receptor, non-parenchymal liver cells

## Abstract

Purinergic signaling has emerged as an important paracrine–autocrine intercellular system that regulates physiological and pathological processes in practically all organs of the body. Although this system has been thoroughly defined since the nineties, recent research has made substantial advances regarding its role in aspects of liver physiology. However, most studies have mainly targeted the entire organ, 70% of which is made up of parenchymal cells or hepatocytes. Because of its physiological role, the liver is exposed to toxic metabolites, such as xenobiotics, drugs, and fatty acids, as well as to pathogens such as viruses and bacteria. Under injury conditions, all cell types within the liver undergo adaptive changes. In this context, the concentration of extracellular ATP has the potential to increase dramatically. Indeed, this purinergic response has not been studied in sufficient detail in non-parenchymal liver cells. In the present review, we systematize the physiopathological adaptations related to the purinergic system in chronic liver diseases of non-parenchymal liver cells, such as hepatic stellate cells, Kupffer cells, sinusoidal endothelial cells, and cholangiocytes. The role played by non-parenchymal liver cells in these circumstances will undoubtedly be strategic in understanding the regenerative activities that support the viability of this organ under stressful conditions.

## 1. Introduction

Although the more recognized tasks in the liver are supported by hepatocytes, non-parenchymal liver cells (NPLCs) play essential roles in the tissular homeostasis and in the structure and maintenance of the organ. For example, hepatic stellate cells (HSCs) support the extracellular matrix production, while Kupffer cells (KCs) play a role in managing infections. Moreover, NPLCs play an essential role in pathological conditions, where they present notable phenotypic adaptations. The objective of the present review was to systematically summarize the experimental evidence on the expression and function of purinergic signaling elements on NPLCs, and its relevance in physiological and pathological conditions. Although the available data are mainly from basic research, we expect that the landscape of NLPC and purinergic signaling opens new ways for new research programs to propose innovative therapeutic targets with clinical impact.

## 2. The Liver and Chronic Disease

The liver is the organ that controls several metabolic pathways that are indispensable in maintaining homeostasis. Some functions of these pathways include urea synthesis, fatty acid metabolism, glycogen storage, albumin, and complement-system component synthesis, among many others [1]. It also coordinates the activity of parenchymal and non-parenchymal cells.

Furthermore, the liver is a complex anatomical structure consisting of hexagonal units (called “lobules”) arranged in a honeycomb-like pattern [2]. Histologically, these units are delimited by vascular points of reference: six portal triads around a central vein. The triad includes branches of the hepatic artery, the portal vein, and the bile duct, and it is connected by a system of liver sinusoids to the central vein. Hepatocytes create cords along the sinusoids, but they differ from those surrounding the central vein in that their metabolic patterns are noticeably different. This cellular arrangement underlies three discrete functional areas termed elsewhere as “liver zonation” [3]. Periportal zone 1 is rich in oxygen and nutrient supply. Its main functions are glycogen metabolism and ammonia handling by urea formation. Intermediate perivenous zone 2 is specialized in xenobiotic metabolism. Perivenous zone 3 is smaller than the others and surrounds the central vein. Glutamine synthetase, increased glycolytic activity, and biotransformative reactions occur in this area. It has been suggested that liver zonation is established as a response of the biochemical gradients of oxygen, hormones, and nutrients implicated in hepatic blood circulation [4], as well as in selective Wnt ligand secretion, acting as morphogens from liver endothelial cells [5].

The liver performs important functions, including absorption, synthesis, storage, and nutrient metabolism. However, it is exposed to toxic metabolites such as xenobiotics, alcohol, and excess fatty acids, as well as to pathogenic factors such as viruses, bacteria, and other microbial components derived from feeding activity. Despite these circumstances, the liver remains in homeostasis thanks to the high plasticity of its cells and the regenerative capacity that allows this organ to continue functioning. For example, mice with a ~70% hepatectomy regain 100% of their size 7 to 10 days after surgery. Under these conditions, total liver regeneration occurs in humans in 3 months [6,7]. However, when there is chronic exposure to toxic agents, this regenerative capacity is insufficient, leading to multiple complications, which trigger liver diseases. The first complication is liver inflammation. If not controlled, liver inflammation generates fibrosis, which can lead to cirrhosis and, eventually, hepatocellular carcinoma [8,9].

### 2.1. Inflammation

When the liver is challenged by metabolic stress due to exposure to toxic agents (xenobiotics, paracetamol, alcohol, fatty acids) or to the presence of pathogens, it promotes an inflammatory response [8]. As a result of the damage, cells release molecules that serve as alarming cellular signals to favor survival and attract immune cells. These are known as danger-associated molecular patterns (DAMPs) and comprise various molecules, such as nucleic acids, nuclear and cytosolic proteins, and nucleotides (mainly ATP) [10]. DAMPs are recognized by pattern recognition receptors (PRR) of resident immune cells of the liver, such as Kupffer cells (KCs), dendritic cells, hepatocytes, and stellate cells (HSCs). Resident immune cells secrete proinflammatory cytokines and chemokines (e.g., IL-1β, IL-6, IL-32, IL-33, and TNF-α [11,12,13]). These cells promote the hepatic extravasation of neutrophils and monocytes, which differentiate into macrophages and magnify the inflammatory response [14]. In addition, ATP, through the P2X7 receptor, can also activate the NLRP3 inflammasome in endothelial sinusoidal cells, hepatocytes, stellate cells, KCs, and neutrophils [15,16]. Stimulation of NLRP3 activates caspase-1, which induces proteolytic processing and the release of interleukins IL-1β and IL-18, further enhancing the proinflammatory response [17]. It has recently been shown that murine neutrophil-specific NLRP3 activation can independently initiate liver inflammation, alter liver development, and promote liver fibrosis [18].

Hepatitis, or liver inflammation, can be beneficial or pathological to the liver, depending on the intensity and duration of the inflammation. When it is controlled, it supports repair, but when inflammation is disproportionate, it becomes pathological.

### 2.2. Fibrosis

In vertebrates, the liver is a soft and well-defined organ. Hepatocytes are the most abundant cells in the liver and are arranged in plates. Among the plates, non-parenchymal cell populations are located in the sinusoids and the Disse space, which lies between the endothelial cells and hepatocytes (Figure 1) [12]. The components of the extracellular matrix (ECM) provide the liver with the appropriate stiffness to fulfill its functions. The ECM is a critical component in the liver, since it provides a surface for cell adhesion, growth, and migration, and serves as a reservoir for signaling molecules [19,20]. When the ECM components build up, and their degradation slows down, fibrosis occurs. Type I collagen, elastin, hyaluronan, fibronectin, and some proteoglycans are primarily involved in this process [21]. Fibrosis causes changes in the structure of the liver and results in the formation of fibrous scars and collagen cords [22].

Although various cell types can alter the ECM, myofibroblasts (MFBs) play a central role in this process, since they are absent from healthy livers [22,23]. Activated HSCs transdifferentiate and generate approximately 90% of MFBs in experimental hepatotoxic models, thus producing profibrogenic factors such as transforming growth factor-beta (TGF-β) and platelet-derived growth factor (PDGF), as well as proinflammatory cytokines [24]. Hence, the stellate cells lose their characteristic basal shape and their vitamin A storages, becoming active producers of type I collagen, α-smooth muscle actin (αSMA), and tissue inhibitor of metalloproteinase 1 (TIMP1) (Figure 1). The accumulation of these components causes phenotypic changes in the liver and reprograms cellular metabolic pathways that affect nutrient exchange [25].

Liver fibrosis may present reversible and irreversible states. Fibrosis is reversible if the promoting etiological agent is removed, as it has been observed in samples from patients with autoimmune inflammatory liver disease, non-alcoholic steatohepatitis (NASH), and alcoholic steatohepatitis (ASH) [22]. Although the reversal of fibrosis is a complex process, macrophages support fibrogenesis by secreting TGF-β and proinflammatory cytokines such as IL-6, IL-1β, and TNF-α, but when there is no inflammatory stimulus, they stop producing proinflammatory molecules and secrete a large amount of collagenase and metalloproteinase enzymes MMP12 and MMP13, which dissolve fibrotic complex aggregates [26]. Furthermore, activated HSCs revert their phenotype in the absence of TGF-β stimulation. Although they do this differently from quiescent cells, HSCs manage to reestablish some of their basic characteristics, such as the accumulation of vitamin A droplets [27]. It has been reported that 50% of these HSCs undergo apoptosis after the cessation of CCl_4_ stimulation [28]. HSCs that revert their phenotype are more susceptible to reactivation by proinflammatory stimuli, as if they had an epigenetic memory [27]. If fibrosis is not reversed, it can damage liver function and increase the risk of developing hepatocellular carcinoma.

### 2.3. Cirrhosis

Currently, cirrhosis is a leading cause of death, with 2 million annually worldwide, accounting for 4% of all deaths [29]. The most common causes of cirrhosis are related to viral hepatitis, alcohol intake, and the spectrum of non-alcoholic fatty liver diseases [29]. Cirrhosis is a consequence of fibrosis. Chronic inflammation causes excessive production of fibrotic scars, which generates plastic and dynamic changes in the stiffness, flexibility, and density of the ECM, thus reducing the parenchymal mass of the liver [30]. Regenerative hepatic nodules replace the typical hepatic architecture; the fenestrations of the liver that allow the exchange of essential molecules are lost, eventually leading to liver failure [31]. In addition, hepatic structural abnormalities prevent proper blood circulation, causing portal hypertension [32], altered bile flow (cholestasis), acidic intestinal pH, as well as changes in the intestinal microbiota and hepatic immune response [33,34]. The chronic inflammation of the liver does not progress to cirrhosis in all patients. However, when it does, the speed at which it occurs can vary from weeks to years. Moreover, cirrhosis often presents asymptomatically [35]. In most cases, symptoms are detected in the advanced stages of the disease, which seriously compromises the survival of the patients.

### 2.4. Hepatocellular Carcinoma

Hepatocellular carcinoma is one of the most lethal and prevalent cancers in the world, according to Global Cancer Observatory (GLOBOCAN). It is usually related to advanced liver disease, with cirrhosis [36] and hepatitis B and C as pathognomonic risk factors for the development of this type of cancer [37]. The mechanisms by which hepatocellular carcinoma is generated are diverse and complex. For example, it is suggested that the continuous renewal of hepatocytes promotes the shortening of telomeres, which generates chromosomal instability [38]. On the other hand, reactive oxygen species (ROS) production and a persistent inflammatory response can be correlated with the activation of genes that favor cell proliferation, thereby sustaining damaged cells that will ultimately lead to cancer [39].

## 3. Purines and Cell Signaling

Chemical communication is a widespread evolutionary strategy to coordinate metabolic and physiological adaptations in a variety of biological systems, which encompass everything, from individual cells to complex organisms. This type of cellular interaction involves the synthesis and release of molecular messengers, as well as the presence of specific receptors in target cells. These receptors are tangled proteins that can undergo concerted conformational changes to allow signal transduction responses. Purines, especially the nucleotide ATP and the nucleoside adenosine (ADO), are ubiquitous adenine-related molecules that play important roles as ligands of an extensive family of membrane receptors. Some relevant physiological processes that involve purinergic signaling are sleep, immunological response, renal electrolyte reabsorption, and platelet aggregation. In addition, it is well accepted that ATP and ADO are valuable mediators in pathological situations such as cancer and inflammation [40,41]. Indeed, the extracellular actions of ATP and ADO as chemical messengers are coordinated with their relevant intracellular roles as enzymatic allosteric factors and energy metabolites [42].

Purinergic signaling is distinctive in one particular aspect: ATP can transform into ADO by consecutive phosphate removal, with ADP and AMP as intermediates in this conversion. Since ATP and ADO recognize their own receptors, the final purinergic communication in a given cellular or tissular system will depend on the expression of selective ATP and ADO receptors, but more importantly, it will depend on the enzymatic machinery of ectonucleotidases and phosphatases, which dictate the proportion of ATP and ADO in the extracellular milieu [43].

Purines are an extensive family of aromatic nitrogen-containing heterocyclic compounds formed by the fusion of pyrimidine and imidazole rings. They are important biochemical, metabolic, and informative entities that function as energy metabolites in the structure of coenzymes and within genetic material. They are also psychoactive factors and chemical messengers. In a cellular context, the two principal purines are adenine (6-aminopurine) and guanine (2-amino-6-hydroxypurine). This review focuses on adenine-related molecules, mainly ATP and ADO, both of which present a ribose as a furanose ring attached to position 9.

ATP is the most abundant adenine nucleotide within cellular systems and is found in the millimolar range. The proportion of the other two nucleotides, ADP and AMP, dictates the equilibrium between anabolic and catabolic reactions [44]. Usually, the extracellular presence of ATP serves to accomplish its messenger/ligand role. For example, ATP can access the extracellular space via exocytosis (secretory vesicles), microvesicles derived from the plasma membrane, transporters (ABC cassettes), pannexin-1 or connexins acting as channels, or purinergic receptor channels such as the P2X7 receptor [45]. However, under cellular stress, ATP flows out of the cells and reaches much higher external concentrations. In this scenario, ATP acts similarly to alarmins or DAMPs, inducing a characteristic inflammatory response [46]. It has also been reported that high levels of ATP are present in the extracellular milieu of cancerous tumors [47].

The appearance of ADO as a ligand in the ECM mainly depends on ATP conversion by a set of dephosphorylation events that remove phosphates to form ADO. ADP and AMP are intermediates in this process. Originally, two ectoenzymes were postulated to dephosphorylate ATP: clusters of differentiation 39 and 73 (CD39 and CD73), which are intrinsic membrane proteins [48]. CD39 turns ATP and ADP into AMP, whereas CD73 converts AMP into ADO. More recently, the family of these enzymes has been expanded to include eight members of the ectonucleoside triphosphate diphosphohydrolase (NTPDase) family, seven mammalian paralogs of the ectonucleotide pyrophosphatases/phosphodiesterases (ENPPase) family, ecto-alkaline phosphatases, and ecto-5′-nucleotidases [41]. The presence of extracellular ADO can end by means of two principal mechanisms: ADO conversion to inosine by the activity of an ecto-adenosine deaminase or ADO internalization to the cytoplasmic milieu by ENT and CNT (equilibrative and concentrative nucleoside transporters, respectively), two types of membrane transporters [49]. 

ATP and ADO interact with a set of specific receptors. These proteins, which are present essentially in every tissue, appeared early in the phylogenetic evolution, since eukaryotic protozoa and simple algae express purinergic receptors [50]. There are two types of receptors for ATP: channel receptors (P2X) and G-protein-coupled receptors (P2Y). P2X receptors are ATP-gated ion channels that are permeable to cations (Na^+^, K^+^, and Ca^2+^). Molecular cloning studies have reported seven genes coding for the P2X1-7 subunits. Each protein is characterized by two transmembrane domains, and the amino and carboxy ends are located on the intracellular side. A functional receptor is a homo- or heterotrimer of P2X subunits [51]. P2Y receptors belong to the family of seven transmembrane regions and influence the levels of second messengers, such as cAMP and Ca^2+^. So far, eight different P2Y receptors have been reported (P2Y1,2,4,6 and 11–14). Apart from ATP, some of these receptors interact with other ligands; for example, ADP with P2YR1, 6, 12–14; UTP with P2YR2 and 4; UDP-glucose with P2YR14 [52]. ADO has four different receptors (A1, A2a, A2b, and A3), all of which belong to the G-protein-coupled receptor family. The signal transduction of ADO receptors usually downregulates the cAMP levels [53].

In summary, purinergic transmission is a recurrent and complex form of cellular communication. Studies have described the ATP and ADO signaling pathways as both complementary and sometimes antagonistic; for example, in some cellular systems, ATP functions as an immunostimulant, while ADO acts as an immunosuppressive agent [54]. Hence, the specific types of ATP and ADO receptors, along with the set of ectonucleotidases in the extracellular space, determine the final purinergic response in a given physiological or pathological event. These parameters, acting in coordination, dictate the proportion of ATP and ADO as ligands and underpin the signaling transduction processes that these purines promote [55].

## 4. Purines in Liver Pathology

Extracellular nucleotides play critical roles in maintaining normal homeostasis and modulate many physiological processes in the liver. However, certain biological stressors alter the release of these nucleotides and the activities of ectonucleotidase enzymes [56,57,58]. In hepatic regeneration, it has been demonstrated that P2Y2 [59], P2X4 [60], and P2X receptors have a global positive impact on the regulation of physiological processes [61]. Additionally, some studies have found that the A3 receptor activation has a protective effect against damage and leads to an increased rate of liver regeneration [62,63]. 

In experimental liver fibrosis, when the damage was induced by chemical toxins such as diethylnitrosamine, carbon tetrachloride (CCl_4_), or thioacetamide, it was demonstrated that P2 [64], P2X7 [65,66], P2Y2 [67], A2, and A1 receptors [68,69,70] are involved in the onset of the fibrotic damage, mediating the activation of various profibrotic factors and the accumulation of collagen, probably through HSC activation. Furthermore, the P2X7 receptor has been shown to mediate the oxidative stress-induced exacerbation of inflammatory liver injury via an NADPH oxidase-dependent mechanism [71]. In addition, a more aggressive effect of hepatotoxic CCl_4_ has also been demonstrated in ectonucleoside triphosphate diphosphohydrolase-2 null mice [72].

Regarding N-acetyl-para-aminophenol (APAP)-induced hepatotoxicity, it was observed that P2X receptors [73], especially the P2X7 receptor, may be involved in necrosis, and that stimulating both of these receptors may enhance cell death independently of immune system activation [74,75,76]. Additionally, the P2Y2 receptor was demonstrated to be involved in increasing the liver damage mediated by APAP and also when hepatitis was induced by the administration of concanavalin A [77].

On the other hand, the P2X4 receptor signaling pathway was responsible for raising the profibrotic profile during bile duct ligation and under a high-fat diet [78]. Furthermore, A1 receptors have been shown to regulate the signaling pathways that lead to fibrosis, as the induction of liver fibrosis associated with CCl_4_ administration was attenuated in mice lacking this receptor [70].

Concerning liver damage induced by lipotoxicity in experimental NASH, it was demonstrated that A2A receptor activation is necessary to induce fibrosis [79]. Moreover, studies have shown that the P2X7 receptor is a key regulator of autophagy induced by metabolic oxidative stress in NASH and therefore, it also modulates hepatic inflammation [80] and the activation of the NLRP3 inflammasome [81,82,83]. In addition, recent studies have proposed that P2X7 receptor activation promotes autophagic flux, thereby improving the outcome of non-alcoholic fatty liver disease (NAFLD) through lipophagy [84].

Moreover, the P2Y2 receptor was shown to play a critical role in the pathogenesis of NAFLD by regulating AMPK [85]. Furthermore, the A2a receptor was demonstrated to halt NASH progression in mice with established steatohepatitis by means of its immunoinhibitory and cytoprotective effects [86,87]. In addition, pharmacological stimulation of the A2a receptor prevents hepatocyte lipotoxicity and inhibits the progression from NAFLD to NASH by interfering with JNK-1/2 activation [79]. On the other hand, the A3 receptor’s pharmacological activation reduced NAFLD by anti-steatotic, anti-inflammatory, and anti-fibrotic actions [88].

According to reports, the P2X4 receptor may boost the progression of alcoholic liver fibrosis by modulating the PI3K/AKT pathway in parenchymal cells or by influencing macrophages to regulate HSC activation [89]. Additionally, the P2X4 receptor promotes the inflammatory response in alcoholic steatohepatitis, while CD39 plays a protective role in regulating P2X4 receptor expression by hydrolyzing ATP [90]. Reports have also indicated that mice lacking the A1a receptor were more susceptible to acute ethanol-induced liver damage than the wild-type mice. This sensitivity was related to increased lipogenesis and lipid peroxidation, as well as to the depletion of superoxide dismutase, which resulted in hepatic steatosis [91].

Regarding the role of purinergic signaling in the activation and recruitment of immune cells in the liver, an increase in CD8+ T-cells was reported in a mouse model of primary sclerosing cholangitis (multidrug resistance protein 2 knockout) with CD39 deletion. This increase exacerbated liver injury and fibrosis, suggesting that extracellular ATP signaling favors CD8+ T-cell infiltration into the damaged region [92]. In agreement, CD39 expression limited the proinflammatory signaling of the P2X7 receptor and cytokine production in a sepsis-induced liver injury model [93]. Additionally, the P2Y2 receptor has also been related to immune response activation in acute liver damage. The P2Y2 receptor −/− mice expressed fewer chemoattractants compared to the wild-type mice in response to concanavalin A, which had a direct impact on hepatic immune cell infiltration [77]. On the other hand, in cholestatic injury, P2X7 receptor activity was reported during NLRP3 inflammasome activation and in the subsequent secretion of potent proinflammatory cytokines [94].

The P2Y2 receptor has been shown to play a fundamental role in hepatocarcinogenesis by stimulating DNA damage responses and hepatocyte proliferation [95]. Moreover, A3 receptor agonists have been utilized to treat hepatocellular carcinoma because of their anti-cancer and hepatoprotective effects [96].

This body of evidence emphasizes how crucial the purinergic system is in regulating different liver diseases. Most of the research has focused on the liver as a whole, without distinguishing between the functional effects observed and the different cells that make up the liver. Generally, such effects and functions are attributed to parenchymal cells; however, some studies have shown that non-parenchymal cells are also essential in the regulation of various pathological conditions of the liver. In this sense, in the following sections, we set out to describe and highlight in detail the importance of the purinergic system in the main non-parenchymal cells of the liver (i.e., HSCs, KCs, LSECs, and cholangiocytes). Note that, although oval cells were considered, no information about purinergic signaling in this cell type is currently available.

## 5. Purinergic Signaling in Non-Parenchymal Liver Cells

### 5.1. Hepatic Stellate Cells

HSCs or Ito cells, are a small population of mesenchymal liver cells whose main structural characteristic is the storage of retinoids, such as vitamin A. In the 19th century, Wilhelm von Kupffer identified these cells using a gold chloride-staining method. HSCs are residents of the perisinusoidal region known as Disse space, located between the hepatocyte cords and endothelial cells [97] (Figure 1). Classical works [98] and recent approaches [99,100,101] support the notion that HSCs represent between 1% and 15% of the total number of liver cells. HSCs are of great biomedical interest because they play a fundamental role in liver fibrogenesis [102,103]. The quiescent HSCs in the normal liver are characterized by cytoplasmic lipid droplets containing retinoids (vitamin A and its metabolites) [97]. These HSCs have a similar genetic program to that of mature adipocytes, such as high expression levels of peroxisome proliferator-activated receptor gamma (PPARγ) and the sterol regulatory element-binding protein 1c (SREBP-1c) [104,105]. Importantly, PPARγ knockdown promotes HSC activation [106].

In response to injury, HSCs undergo a transdifferentiation process known as “activation”, whereby they lose the lipocyte phenotype and acquire MFB characteristics [107]. Cells with the MFB phenotype lose their lipid droplets and express αSMA and collagen 1a1 (Col1a1), a pivotal step for fibrosis induction. Recent studies support the idea that HSCs are the main source of MFB in the fibrotic liver [108,109]. When HSCs are cultured in plastic Petri dishes, they go through spontaneous and progressive activation within 5 to 7 days [97].

Interestingly, evidence suggests that HSC activation is reversible. In hepatocellular damage induced by CCl_4_, the removal of the hepatotoxic stimulus is enough to trigger the reversion of a fraction of the fibrotic MFB phenotype, by downregulating fibrotic genes and inhibiting apoptotic death [27]. Furthermore, the administration of PPARγ or exogenous expression of SREBP-1c have also been reported to revert the MFB phenotype to quiescent HSCs [104,105]. Novel therapeutic approaches to treat hepatic fibrosis are made possible by the reversal mechanisms of the HSC-activated phenotype. In the next section, we will review the direct actions of purines on HSCs mediated by purinergic receptors or ectonucleotidase activities.

#### 5.1.1. Nucleotide Signaling in Hepatic Stellate Cells

A fundamental study conducted on rat HSCs before the cloning of purinergic receptors described the production of IP_3_—the intermediate metabolite that mobilizes intracellular Ca^2+^—and the induction of cell contraction in response to purinergic substances. Stimulation with ATP, UTP, ADP, or 5′-O-(3-thiotriphosphate) incremented intracellular levels of inositol phosphates and cytosolic Ca^2+^ ([Ca^2+^]_c_) and induced cellular contraction [110]. These observations suggested that purines regulate HSC physiology through their action on specific receptors (Figure 2, Table 1).

In 2004, Rebeca Wells and collaborators characterized the expression and function of P2YRs in quiescent and activated rat HSCs. Quiescent HSCs express transcripts coding for P2Y2 and P2Y4 receptors, whereas activated HSCs express P2Y1 and P2Y6 receptors. Pharmacological characterization of [Ca^2+^]_c_ mobilization assays mediated by purinergic agonists confirmed these findings. These findings show that, in addition to the P2Y6 receptor, activated HSCs express an ATP-sensitive receptor that is not a P2Y1, P2Y2, or P2Y4 receptor.

Importantly, the stimulation of activated HSCs with UDP increased procollagen-1 expression levels threefold, suggesting a pro-fibrotic role for the P2Y6 receptor [111]. On the other hand, in a model of alcoholic liver toxicity, administering acetaldehyde to rat quiescent HSCs for 48 h increased the expression of the P2X7 receptor, along with the fibrotic markers αSMA and Col1a1. Furthermore, P2X7 receptor activation with 2′(3′)-O-(4-Benzoylbenzoyl) adenosine-5′-triphosphate (BzATP) in quiescent HSC potentiated acetaldehyde-dependent proliferation, cyclin D1 expression, and the upregulation of inflammation and activation markers. Conversely, knocking down the P2X7 receptor expression or incubation with the antagonist A438079 inhibited acetaldehyde-dependent HSC activation [112], suggesting a role for the P2X7 receptor and extracellular ATP in rat HSC activation. The P2X4 receptor, for its part, is overexpressed in the liver of mice with fibrosis induced by hepatotoxic damage (ethanol plus CCl_4_), as well as in the immortalized cell line HSC-T6, in response to acetaldehyde administration. Receptor blocking with 5-(3-Bromophenyl)-1,3-dihydro-2*H*-benzofuro[3,2-*e*]-1,4-diazepin-2-one (5-BDBD) prevented the activation of HSC-T6 cells [89].

A study on the contribution of DAMPs and their receptors to hepatic fibrogenesis found that the P2Y14 receptor is enriched in HSCs. Accordingly, their ligands (UDP-glucose and UDP-galactose) are released by dying hepatocytes. An robust ligand–receptor interaction analysis, combining proteomics and single-cell transcriptomics, confirmed the relationship between both components. Importantly, the stimulation of the P2Y14 receptor by specific agonists or co-culture with dying hepatocytes activated HSCs through a pathway involving extracellular signal-regulated kinase (ERK) and Yes-associated protein (YAP). Moreover, the systemic or HSC-specific P2Y14 receptor abrogation diminished fibrosis induction in different models of liver damage [113].

#### 5.1.2. Ectonucleotidases in Hepatic Stellate Cells

Regarding ectonucleotidase expression and function in HSCs (Table 1), an early study determined that HSC activation is associated with NTPDase2 expression [111]. The NTPDase family has also been analyzed in other studies. HSCs cultured for 1, 4, or 7 days expressed the transcripts coding for NTPDase1 (CD39) and NTPDase2. The comparative analysis of ectonucleotidase activity between quiescent (1 day) and activated (7 days) HSCs showed that both types can hydrolyze ATP in ADP and AMP, suggesting that NTPDase1 is expressed similarly in both conditions. In hepatic fibrosis induced by the administration of CCl_4_ for 6 weeks, NTPDase2 co-localized with activated HSC [111]. Recently, it has been shown that the pharmacological blocking or genetic inhibition of CD39 expression in the T6 rat HSC line abolished acetaldehyde-dependent proliferation and expression of fibrosis markers [114]. The ENPP family (ENPPs: NPP2-3) was also detected in the mouse GRX HSC line. ENPP activity was higher in quiescent HSCs compared to activated HSCs, and during activation, the NPP2 expression increased, whereas the NPP3 expression decreased [115].

CD73 (5′-NT) is the enzyme that catalyzes the conversion of AMP into ADO. Interestingly, CD73 knockout mice are resistant to experimental fibrosis, indicating that ADO plays a part in fibrosis induction [116]. Pivotal studies in rat HSC primary cultures have shown that the CD73 expression is very low in the quiescent phenotype. However, the activation process triggers a strong increase in the expression levels of this enzyme, which is consistent with activated HSCs. Similar findings were made in immortalized and primary cultured human HSCs. Furthermore, it was determined that CD73 overexpression is transcriptionally regulated by the canonical TGF-β pathway via the SP1 and SMAD response elements in the *NT5E* promoter [117].

Other reports have supported a role for CD73 in HSC activation [118] mainly using alcohol-related liver fibrosis models [119]. Thus, silencing CD73 in the HSC-T6 line blocked the acetaldehyde-dependent accumulation of αSMA and Col1a1 and induced apoptotic cell death and cell cycle arrest [119]. Similarly, the expression of fibrotic markers and cell proliferation was inhibited by knocking down the ectonucleotidase with a siRNA or by the inhibitor APCP in LX2 and HSC-T6 lines [120]. In agreement, the overexpression of CD73 potentiates acetaldehyde-dependent fibrosis [119]. Between the possible mechanisms explaining the findings described above, it has been proposed that alcoholic liver damage potentiates autophagy induction in activated HSC in a CD73-dependent way by an AMPK/AKT/mTOR pathway [121]. CD73 regulates HSC senescence throughout the p53 pathway by regulating aurora kinase A (AURKA) expression [119]. In addition, a role for the Wnt/β-catenin pathway has been suggested in the execution of the HSC activation program [118].

#### 5.1.3. Adenosine Signaling in Hepatic Stellate Cells

Adenosinergic signaling has been investigated in HSCs (Table 1). Diverse evidence highlights the importance of A2a-mediated ADO signaling in these cells [122]. Thus, in the human HSC line LX2, the activation of the A2a receptor inhibited the PDGF-induced chemotactic response by modulating a pathway that involves adenylate cyclase and cAMP. This led to the upregulation of TGF-β and collagen I RNA production [122]. In this context, it was reported that the A2a-dependent collagen type I and type III overexpression is mediated by a pathway that involves protein kinase A (PKA), src kinase, and mitogen-activated protein kinases ERK and p38 [123], suggesting a role for the ADO/A2a pathway in fibrosis onset. The latter effect can be modulated by interferon γ (IFN-γ), which downregulates adenylate cyclase expression and inhibits A2a-dependent signaling [124].

In addition, in LX2 cells, it was described that the A2a receptor mediates the loss of actin stress fibers via a pathway that comprises cAMP, PKA, and Rho inhibition and suppresses endothelin-1 and lysophosphatidic acid-induced cell contraction [125], suggesting a role in the control of mesenchymal phenotype acquisition. In the same model and rat primary HSCs, A2a activation with NECA or CGS21680 increased cell proliferation and inhibited senescence entry; these changes were accompanied by the downregulation of p53, Rb, and Rac1 and by the inhibition of ERK phosphorylation. These effects were blocked by the A2a antagonist ZM241385 and a PKA inhibitor and reproduced by a cAMP analog and by forskolin, an adenylate cyclase activator [126]. ADO was unable to induce increments in [Ca^2+^]_c_ in LX2 cells. However, the nucleoside prevented Ca^2+^ mobilization induced by ATP or PDGF [122].

Interestingly, the A2a receptor antagonist caffeine and ZM241385 inhibited the acetaldehyde-dependent expression of fibrotic markers in HSC-T6 cells. The pathway regulated by A2a involved cAMP, PKA, Src, and ERK in pro-collagen I regulation and P38 in collagen III regulation [127]. On the other hand, caffeine inhibited the activation of primary mice HSCs acting on ADO receptors; however, in this model, the mechanisms were independent of intracellular Ca^2+^, phosphodiesterase activity, cAMP levels, and ERK phosphorylation, but they relied on Akt1 signaling [128]. In agreement, in systemic models of alcoholic liver disease in rats, caffeine administration decreased fibrosis markers induced by alcohol in isolated HSCs [130].

It was also demonstrated that ADO and the aspartate salt of ADO (IFC305 compound) reversed CCl_4_-induced hepatic fibrosis and cirrhosis in rats. IFC305 prevented the activation of primary rat HSCs and inhibited the proliferative effect of PDGF on HSCs. However, the participation of ADO receptors in this protocol has yet to be elucidated [129].

### 5.2. Kupffer Cells

KCs are the resident macrophages of the liver; they are located in the Disse space, surrounding the sinusoid. KCs were first described in 1873 by von Kupffer and constitute approximately 15% of the liver cell population, 30–35% of the non-parenchymal cells [131], and 95% of the macrophages in the mammalian body [132]. KCs are an essential component of innate immunity, acting as the first line of response against potentially harmful elements [133]. They are also the first macrophage population to come in contact with microbes, endotoxins, and microbial debris from the gastrointestinal tract [133]. KCs originate from bone marrow-derived monocytes that migrate to various organs to differentiate into macrophages [134]. In adult organs, they can originate from stem cells.

KCs are differentially distributed throughout the liver; 43% are located in the periportal zone 1, 28% in the intermedial perivenous zone 2, and 29% in zone 3 [135]. Moreover, KCs are the first to be exposed to substances absorbed by the intestinal tract and, consequently, have been considered scavengers of the liver for their ability to eliminate microbes, endotoxins, dead cells, and xenobiotics [136]. Evidence demonstrates that KCs can also function as antigen-presenting and essential regulators of the liver’s inflammatory response. This has various pathophysiological implications, including the fact that they are able to modulate tumor growth and fibrosis onset.

#### Nucleotide Signaling in Kupffer Cells

The role of purinergic receptors has been widely described in circulating and resident macrophages. Monocytes or macrophages express all the P2Y receptor subtypes [137]. An early study identified the P2Y2 receptor as the main purinergic receptor mediating nucleotide-dependent Ca^2+^ signaling in peritoneal macrophages. After activation with LPS, the P2Y6 receptor elicited Ca^2+^ mobilization from the endoplasmic reticulum [138]. In macrophages, P2Y receptors mediate proinflammatory or anti-inflammatory responses [137]. For example, the macrophage-expressed P2Y2 receptor detects nucleotides released by dying cells, enabling their phagocytosis and thereby maintaining tissue homeostasis [139]. In human macrophages, the P2Y2 receptor and the P2Y12 receptor cooperate to kill and engulf bacteria [140]. In the peritoneal macrophages, the P2Y2 receptor mediates the release of proinflammatory mediators, such as IL-1β, in response to LPS stimulation [141].

The P2X7 receptor, for its part, is the best characterized receptor in macrophages [142]. Its expression is upregulated during the differentiation from monocytes to macrophages. The protein level of P2X7 and its ability to induce megapore formation is greater (tenfold) in macrophages compared to monocytes, but it is clearly expressed in both phenotypes [143,144]. In macrophages, P2X7 receptor activation induces ion currents and membrane depolarization [143,144], large pore activation [145], the secretion of inflammatory mediators such as IL-1β [146,147], NLRP3 inflammasome activation [146,147,148,149], and pyroptosis [146].

Purinergic receptor expression and function have also been explicitly characterized in KCs (Table 2). An early study using immunofluorescence to analyze the expression of P2X receptor subunits in KCs (ED1^+^) identified P2X4 receptor and P2X6 receptor subunits. In the same study, intraperitoneal injection of LPS upregulated the P2X6 receptor expression [150]. The P2X7 receptor has also been found to induce inflammation in response to diverse stimuli. In isolated KCs, extracellular ATP acts on the P2X7 receptor, leading to an NLRP3 inflammasome assembly and, consequently, IL-1β release [76]. In mouse primary KCs, treatment with the environmental contaminant imidacloprid, a compound that causes liver toxicity, induced P2X7 receptor-dependent pyroptosis [151].

Similar observations have been made in immortalized KC lines. In the mouse KUP5 cell line primed with LPS, stimulation with 3 mM of ATP induced channel aperture, the formation of large pores, and MAPK activation, all of which are hallmarks of the P2X7 receptor activation [152]. Furthermore, the P2X7 receptor activation also induced the release of inflammatory messengers, such as prostaglandin E2, IL-1β, and high mobility group box-1 protein (HMGB1) [152]. In KUP5 cells, whether primed with LPS or not, exposure to silica nanoparticles (30 nm diameter) induced ATP release, activating the P2X7 receptor and eliciting IL-1ẞ production [153].

On the other hand, a study reported that the P2Y6 receptor is overexpressed in the liver, specifically in the KCs of mice with alcohol-induced steatohepatitis. In the liver of these animals, the administration of UDP, a P2Y6 receptor agonist, elevated the levels of damage markers and the release of proinflammatory cytokines TNF-α, IL-1β, and IL-6 induced by alcohol treatment. However, administering MRS2678, an antagonist of this receptor, counteracted these effects. The study also showed that the P2Y6 receptor is functional in the mouse macrophage line RAW264.7, and that blocking its activity attenuated the ethanol-dependent inflammatory response [154].

The ADO receptor expression in KCs was reported in the context of liver grafts. Reperfusion of the organ induces the release of TNF-α by KCs, affecting graft function and causing rejection. However, ADO administration in ischemic preconditioning, which involves a transitory occlusion of the portal triad followed by reperfusion prior to ischemia, protects graft viability [155,156]. In cultured rat KCs, ADO inhibited LPS-dependent TNF release. This effect was also reproduced by the A2 receptor agonists 5-N-ethyl carboxamido adenosine, 2-chloro-adenosine, and R-phenylisopropyl adenosine. CGS15943A, a selective antagonist of the A2 receptor, reverses the inhibition exerted by the aforementioned agonists. Furthermore, the suppression of TNF alpha was also reproduced by dibutyryl-cAMP [157]. According to a recent study, A2bR expression increased in liver homogenates and primary cultures of KCs in alcohol-fed mice, indicating that A2b plays a role in the inflammatory response that underlies alcoholic hepatitis. Similar results were obtained with the human macrophage line RAW264.7 in response to ethanol. Knocking down the A2b receptor in RAW264.7 cells promoted an increment in the release of inflammatory cytokines (IL-6, IL-1β, and TNF-α). In agreement, the A2b receptor overexpression results in an attenuation of the inflammatory response [158]. These results suggest that ADO, acting through the A2b receptor, modulates the inflammatory response in KCs in alcoholic steatosis.

The concentration and proportion of nucleotides and ADO in the extracellular space are critical for regulating inflammation in tissue damage conditions, such as steatosis and fibrosis. Ectonucleotidases are critical elements of these mechanisms. CD39 is one of the main enzymes regulating ATP concentration in the extracellular space. Systemic or myeloid cell-specific deletion of this enzyme increased the sensibility of mice to pharmacologically induced biliary fibrosis, as well as the expression levels of the transcripts for inflammation markers (i.e., TGF-β1 and TNF-α) [159]. CD39^−/−^ mice also show greater inflammation and liver damage in response to sepsis, probably due to macrophage activity by deficient extracellular ATP scavenging [93].

### 5.3. Purinergic Signaling in Liver Sinusoidal Endothelial Cells

Liver sinusoidal endothelial cells (LSECs) are highly specialized vascular cells surrounding the hepatic lobule [160,161]. They have pores, called fenestrae, with diameters that range from 50 to 300 nm. Fenestrae travel along the cytoplasm and form clusters called “sieve plates”. These pores regulate the solute flux towards the hepatic lobule. The fenestration in sieve plates has distinctive patterns according to the zone of the lobule. The periportal fenestrae are large and scarce, the centrilobular ones are small and abundant, and the pericentral LSECs are few and scattered. LSECs are different from other endothelial cells in that they lack a basal membrane and thus allow solutes to move along the space of Disse and into the sinusoidal circulation. LSECs also have high endocytic capacity with specific receptors, functioning as scavengers for several types of collagen, hyaluronan, chondroitin sulfate, oxidized low-density lipoproteins, and immune complexes formed with IgG [162,163,164,165].

Purinergic signaling was first explored in vascular endothelial cells before the entire family of P2 receptors was fully described. Subsequent studies showed that ATP and ADP stimulate the release of prostacyclin (PGI2) and nitric oxide (NO), both of which inhibit platelet aggregation via a mechanism involving Ca^2+^ mobilization. These studies helped to demonstrate that extracellular nucleotides modulate intravascular platelet aggregation when acting through their receptors [166,167]. In LSECS (Table 3), it was shown that prostaglandin E2 (PGE2) is the main prostanoid secreted by primary cultures in response to extracellular nucleotides. Pharmacological approaches suggested that this response was mediated by P2YR and ADO receptors [168].

CD39 is the most expressed ectonucleotidase in the vascular endothelium, and it has been demonstrated that this enzyme is a regulator of platelet activation [172]. CD39 null mice show alterations in blood coagulation and angiogenesis [173]. In the liver, partial hepatectomy induces a specific increment in the expression level of CD39 in LSECs, which was detected from 6 h to 5 days’ post-surgery. In CD39^−/−^ mice, hepatic regeneration was impaired, and it was observed that throughout the regenerative process, angiogenesis decreased, while hepatic damage increased. LSECs exhibited a decrease in HGF release and an attenuation in the activation of VEGF receptor 2. The latter was attributed to a lack of extracellular nucleotide transactivation [169]. Interestingly, a co-culture of mouse LSECs expressing CD39, but not LSECs from CD39^−/−^ mice, boosted the proliferation of tumor cells (B16/F10 from melanoma) and limited the induction of extracellular ATP-dependent cell death [170].

Activation of the A2a receptor with the agonist CGS21680 protected LSECs from the damage induced by ischemia reperfusion, favoring the expression of genes associated with cytoprotection, regeneration, energy metabolism, and response to oxidative stress [171].

### 5.4. Purinergic Signaling in Cholangiocytes

Cholangiocytes, or biliary epithelial cells, are specialized cells forming the biliary epithelium and lining the bile ducts. Although they represent less than 5% of the nuclear mass of the liver, they form an extensive branching network and are thought to account for up to 40% of the bile volume in humans [174].

Cholangiocytes are polarized cells with apical and basal membranes that maintain bile flow via the cilium system; they also maintain intraductal homeostasis by active biomolecule transport; modify bile by secreting bicarbonate (HCO_3_^−^) through the plasma membrane domain; maintain cross-ductal interaction in the liver, depending on their tight junctions and immunoglobulin A (IgA) secretion; and reabsorb different molecules, including bile salts and acids, glucose, amino acids, and ions [175,176,177]. Cholangiocytes are classified as small or large, according to their morphology [178]. Large cholangiocytes line the larger branches of the biliary tree and form more complex structures than those of small cholangiocytes. In addition, large cholangiocytes engage in hormone-modulated bile secretion, while small cholangiocytes can proliferate and exhibit functional plasticity during diseases [179,180,181]. Small cholangiocytes are capable of self-replication during liver injury, confirming their potential in liver regeneration and ductular reaction [182]. Ductular reaction is a complex of dynamic interactions among liver parenchymal cells, stromal cells, and immune cells, which serves a crucial machinery during liver injury–regeneration, fibrogenesis, and malignant transformation processes [183].

Cholangiocytes have crucial functions in liver pathobiology. They interact with both intra- and extrahepatic ductal environments and are exposed to both hepatic- and gut-derived stimuli, such as pathogen-associated molecular patterns (PAMPS), DAMPs, and microorganisms) [184,185]. As a result, cholangiocytes are collateral targets of various liver disorders, such as NAFLD, NASH, and alcohol-related liver disease. Cholangiocytes are also directly injured in chronic cholestatic liver diseases, including primary biliary cholangitis, primary sclerosing cholangitis, biliary atresia, and cholangiocarcinoma (CCA) [186,187]. Furthermore, stimulated cholangiocytes can adopt varying secretory phenotypes. Some findings suggest that the bile duct-derived ductular cells play active roles in liver regeneration. Interestingly, activated cholangiocytes exhibit a peculiar secretory phenotype that shapes their surrounding microenvironment by modulating immune cell recruitment and mesenchymal cell migration and activation [188]. The release of cholangiokines (cholangiocyte-secreted cytokines, including chemokines, growth factors, and others) is associated with cellular responses, which are affected by tissue inflammation, infection, and metabolic dysregulations. Both acute and chronic liver disorders have been shown to alter the secretory profiles of cholangiocytes [189,190,191].

#### 5.4.1. Nucleotide Signaling in Cholangiocytes

Cholangiocytes exhibit constitutive ATP efflux. In the monolayers of rat cholangiocytes, it was demonstrated that ATP release is polarized with apical concentrations (lumenal) of approximately 250 nM, which are fivefold greater than basolateral concentrations (approximately 50 nM) [192]. These values are within the range required for the half-maximal stimulation of Cl^−^ secretion in biliary epithelium. Accordingly, the ATP in bile is most likely derived from release across the canalicular membrane of hepatocytes and the apical membrane of cholangiocytes. The most potent stimulator of hepatic ATP release is an increase in cell volume. Moreover, exposure to exogenous ATP causes a rapid decrease in cell volume due to K^+^ and Cl^−^ channel activation. These and other findings suggest that extracellular ATP plays a key role in recovery from cell swelling in different liver and biliary cell models [193,194,195].

Additionally, cholangiocytes express MDR-1 and CFTR ATP-transporting proteins [196]. By modulating the function of these proteins, it is possible to regulate ATP release, and recent studies indicate that these transporters are more likely to serve as ATP channel regulators, increasing the amount of ATP released through an as-yet-unidentified ATP channel [197].

Furthermore, increases in cell volume cause activation of phosphoinositide 3-kinase (PI 3-kinase) [198], which plays a central role in the regulation of hepatocyte and cholangiocyte ATP release [199,200]. PI3-kinase signaling could lead to the insertion of new ATP channels or, alternatively, could directly modulate the opening of pre-existing ATP channels through the generation of secondary messengers.

On the other hand, the degradation of extracellular nucleotides is an essential step in the overall process of purinergic signaling. In vitro studies of polarized biliary epithelia demonstrated substantial differences in the kinetics of ATP degradation between apical and basolateral surfaces. In the apical compartment, the time course of nucleotide degradation can be described by a single exponential, resulting in the clearance of 13% of ATP within the first minute. In the basolateral compartment, the time course was more complex, suggesting the presence of more than one degradation pathway. These differences in degradation kinetics are likely because of location-specific biliary ecto-ATPases [192].

Current research is aiming to fully elucidate the roles of ADO, P2X, and P2Y receptor signaling in hepatic and biliary function. Some functions have been demonstrated in cell volume regulation, paracrine signaling, and bile formation.

The maintenance of cell volume is crucial for liver cells that receive dual perfusion from both the systemic and portal circulations and are exposed to unusually large changes in the concentrations of solutes, such as amino acids, bile acids, and glucose, between the fasted and fed states. Extracellular ATP plays a key role in restoring the cell volume to basal values in a process referred to as regulatory volume decrease [193,195].

In the liver, ATP release appears to represent a paracrine signal that is responsible for the coordination of calcium signaling among cells, suggesting a role in cell-to-cell coupling along the sinusoid. Specifically, the ATP released by one cell binds to P2 receptors on neighboring cells, stimulating a rise in cytosolic calcium [201]. Simultaneously, ATP released from the canalicular hepatocyte membrane or the apical cholangiocyte membrane may significantly contribute to bile formation by stimulating the apical purinergic receptors.

ATP is released into bile by both hepatocytes and cholangiocytes, and functions as a potent autocrine/paracrine stimulus for cholangiocyte secretion [202,203]. When the cell volume increases, the ATP membrane permeability is enhanced by PKC activation, resulting in the elevation of extracellular ATP [204]. Once in the bile, ATP has direct access to the apical cholangiocyte membrane, where it induces increases in intracellular Ca^2+^ concentration by purinergic receptor activation and activates the membrane Cl^−^ and K^+^ channels [205]. The resulting transepithelial transport of Cl^−^ significantly contributes to the alkalinization and dilution of the bile [206]. Thus, the description of proteins implicated in cholangiocyte secretion by purines represents an excellent tool for understanding the entire mechanism implicated in the purinergic regulation of bile physiology and pathophysiology.

P2YRs are recognized as the prototype effector pathway responsible for Ca^2+^-dependent secretory responses [205,206]. ATP binding to these G-protein-coupled receptors stimulates phospholipase C, generates IP_3_, and releases calcium from intracellular stores. The P2Y2 receptor represents the best characterized member of this family. cDNAs, mRNAs, and protein corresponding to the P2Y2 receptor are readily detectable in cholangiocytes, and the P2Y2 receptor-preferring agonist UTP effectively stimulates cholangiocyte secretion, which is consistent with an important physiological role for this receptor [205,206].

The P2Y12 receptor is also localized in the apical membrane of cholangiocytes [192,205,207]. This receptor appears to be restricted to the cilium, a mechanosensory, osmosensory, and chemosensory organelle that detects and transmits different stimuli from the bile into the cell, modifying cellular functions [208,209]. P2 receptors have also been observed, although less frequently, at the basolateral domain of cholangiocytes [206,210]. By acting on these P2 receptors, ATP triggers an increase in intracellular Ca^2+^, which results in the activation of K^+^ and Cl^−^ channels [211].

On the other hand, transcripts for P2XR 2, 3, 4, and 6 have been detected in cultured rat cholangiocytes and cholangiocyte lysates. The P2X4 receptor protein was readily detected, and immunohistochemical staining of intact rat liver revealed that the P2X4 receptor concentrated in intrahepatic bile ducts. The functional significance of the P2X4 receptor was assessed in isolated Mz-ChA-1 cells using the P2X4 receptor-preferring agonist BzATP. Moreover, BzATP elicited robust Cl^−^ secretory responses, demonstrating that the P2X4 receptor is an important component of the purinergic signaling complex that modulates biliary secretion [210] (Table 4).

#### 5.4.2. Adenosine Signaling in Cholangiocytes

The functional expression of ADO receptors and transporters has been characterized in polarized rat cholangiocytes. Studies demonstrated that biliary cells exclusively express the efficient ADO transporters CNT3 (located only in the apical membrane) and CNT2 (located in the apical and basolateral domains). In both domains, cholangiocytes also express the high-affinity ADO receptor A2a, whose activation may modulate the activity of apical CNT3 in a domain-specific manner. The A2a receptor-mediated regulation of CNT3 is dependent upon the cAMP/PKA/ERK/CREB axis, intracellular trafficking mechanisms, and AMPK phosphorylation. Moreover, extracellular ATP (an ADO precursor) is able to exert an inhibitory effect on the apical activity of both CNT3 and CNT2, which seemingly involves different P2 receptors [212].

Purinergic signaling has also been investigated in CCA, a rare malignancy of the epithelial cells from various areas of the biliary tree, including intrahepatic, perihilar, and extrahepatic bile ducts. Studies demonstrated that CCA cell lines and MMNK-1, an immortalized CCA cell line, share P2Y2 and P2X3 receptors. On the other hand, P2Y6, P2Y13, and P2X7 receptors were expressed in CCA cells but not in immortalized CCA cells. MMNK-1 cells showed some resistance to cell proliferation and cell motility inhibition by ATP and ADO. Therefore, it is suggested that the receptors expressed in CCA cells but not in MMNK-1 cells might be responsible for the aforementioned inhibition. In addition, ADO was shown to suppress cell proliferation and motility [213,214]. However, further studies are required in order to examine the mechanism of inhibition caused by ADO on CCA cells.

## 6. Therapeutic Targeting of Purinergic Signaling in Non-Parenchymal Liver Cells

Taking advantage of the abundant knowledge accumulated regarding the importance of ATP and ADO signaling in diverse tissues, as well as the reported 3D structures of a variety of purinergic receptors, research groups in recent years have postulated the translational use of drugs that interact through elements of purinergic signaling. For example, a number of purinergic drugs are commercially available, including P2Y12 receptor antagonists for stroke and thrombosis, P2Y2 receptor agonists for dry eye, A1 receptor agonists for supraventricular tachycardia, and P2X3 receptor antagonists for the treatment of chronic cough, visceral pain, and hypertension [215,216].

As for liver pathologies, some promising therapeutic advances have been reported [58]. For example, MRS1754, an A2B ADO receptor antagonist, has shown positive actions as an antifibrotic agent [58,217]; A438079, a P2X7R antagonist, promoted a reduction in inflammation and collagen accumulation [65]; 5-BDBD, a P2XR antagonist, alleviated liver fibrosis in an experimental model of methionine-choline deficient diet [78]; NF340, a P2Y11R antagonist, counteracted HCC cells [78,218]; CF102, an A3AR agonist, reduced HCC cell availability [219].

Among these studies, one well-characterized effect is the targeting of A3 receptors to address non-alcoholic fatty liver disease/non-alcoholic steatohepatitis (NAFLD/NASH). The expression of the A3 receptor in NAFLD patients is significantly lower than in healthy persons; in addition, a high fat diet in mice lacking A3-receptor expression results in an enhanced expression of genes involved in hepatic inflammation and steatosis [220]. Moreover, the administration of the A3-receptor agonist prodrug MRS7476 protects against NASH development in STAM mice [221]. Another A3-receptor agonist, C1-IBMECA (namodenoson), was tested in the treatment of NASH in mice. Namodenoson is currently in a Phase 2 clinical trial for NASH therapeutics (ClinicalTrials.gov Identifiers: NCT02927314 and NCT04697810, accessed on 31 May 2021) [88]. Moreover, another A3 agonist, named CF102, was administered to patients with advanced HCC in a phase I/II trial. It was observed that this drug is safe, has a linear pharmacokinetics, and reverses the metastatic process, demonstrating clinical applicability [222].

Another case is IFC305, an analog of adenosine, which is able to reverse fibrosis in pre-established rat cirrhosis [223]. Mechanistic studies demonstrated that IFC305 prevents HSC activation [129] and modulates the inflammatory response dependent on Kupffer cells [224]. Clinical trials and pharmaceutical developments are currently in development [225].

Finally, the scarcity of reports regarding the purinergic system in well-recognized experimental models of hepatic disorders is notable. For example, data in Pubmed (August, 2024) did not show any record of purinergic signaling and liver injury in relation to hemochromatosis, Wilson disease, and herbal-induced hepatitis, whereas from among 47,993 citations for drug-induced liver injury, only 18 (0.037%) were related to elements of ATP/ADO signal transduction. Indeed, this presents an opportunity to encourage basic and clinical groups to continue the biomedical research on liver diseases and cellular purinergic communication in the near future.

An interesting parallelism exists between the liver and kidneys from the perspective of purinergic physiopathology. Both organs are highly dependent on nutrient processing, they are formed by the coordinated function of different cellular components, and the hepatic and renal responses to ATP are proinflammatory, whereas ADO acts as an anti-inflammatory factor [226]. In this context, it has been recognized that the presence of P2X signaling cascades play a role in acute kidney injury [226,227], and that the dysregulation of the P2 receptor in urologic diseases, as well as the participation of purinergic receptors in urinary infections by bacteria [228] and kidney transplantation, also play a role [229].

## 7. Concluding Remarks

The liver is a complex system. It is formed by a variety of cellular types that harmonize their operation to achieve the dynamic adaptation needed for a healthy hepatic physiology. In this context, efficient communication and coordination between the parenchymal and non-parenchymal cells is imperative to ensure a correct metabolic and protective hepatic response. Among the multiple cellular messengers that exist within the liver, this review focused on how hepatic ATP and ADO signaling during fibrotic, cirrhotic, and cancerous damage affects HSCs, KCs, biliary cells, and LSECs.

Indeed, this approach is necessary to complement the literature on purinergic function in hepatocytes and ultimately achieve an integrated overview of the liver in both healthy and pathological conditions.

## Figures and Tables

**Figure 1 ijms-25-09447-f001:**
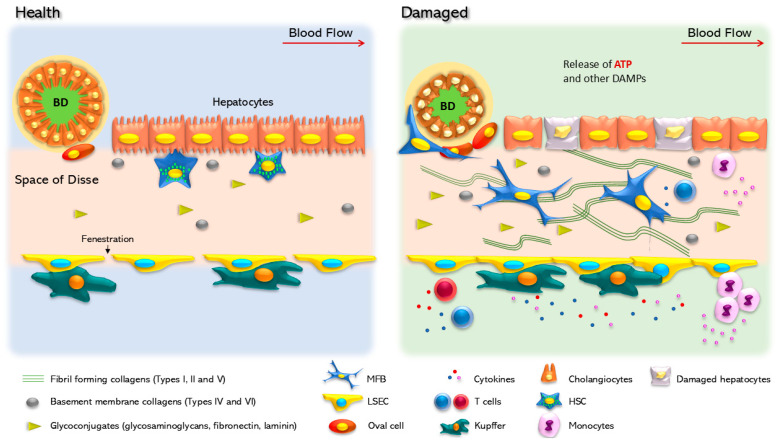
Cellular changes in healthy and damaged liver. In the healthy liver, parenchymal cells are well organized, following a portal/central zonation in basal contact with the extracellular matrix. In this condition, hepatocytes show cilium and nuclear integrity. Hepatic stellate cells (HSCs), located in the Disse space, between the parenchymal layer and the liver sinusoidal endothelial cells (LSECs), show a quiescent phenotype characterized by vitamin A (retinol) storage. LSEC surrounds the Disse space and in specific zones form specialized structures named fenestra that regulate the flux of metabolites to the acinus. Kupffer cells are in contact with LSECs in an inactivated state, and oval cells are in a quiescent state surrounding the biliary duct (BD), which is made up of cholangiocytes. In response to a chronic injury, the liver suffers continuous scarring, until fibrosis is established as a result of the modification and accumulation of the extracellular matrix. Hepatocytes are the first cells to respond and adapt to stressful stimuli. Non-parenchymal cells undergo a set of phenotypic changes. Experimental HSC activation is a transdifferentiation process that promotes HSC conversion into myofibroblasts, a phenotype characterized by the production of extracellular matrix components, mainly collagen Iα. Kupffer cells are also activated and initiate the secretion of proinflammatory cytokines. In addition, LSEC loses the fenestra and releases inhibitors of platelet aggregation, such as nitric oxide and prostaglandin G12 (NO and PGI2). Cholangiocytes within the biliary duct lose the cilia and change their secretory profile.

**Figure 2 ijms-25-09447-f002:**
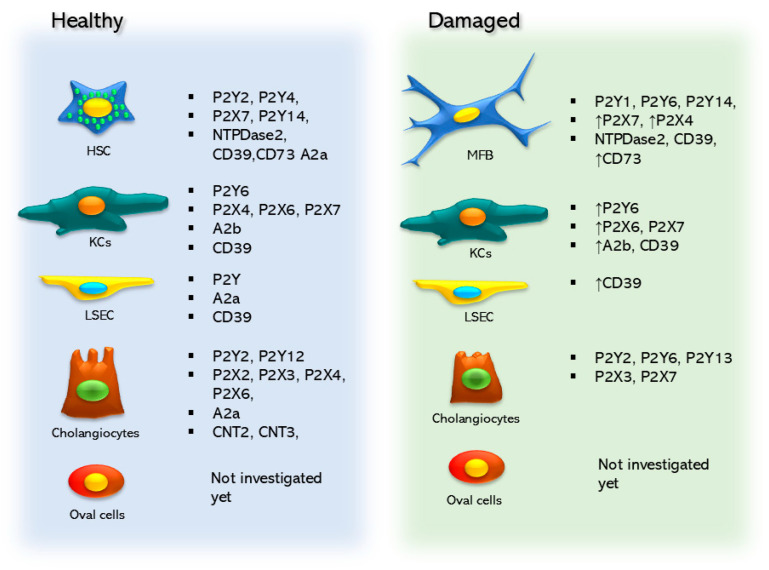
Purinergic elements are expressed in the non-parenchymal cell types of the liver, and these elements change and are modified between healthy and damaged states.

**Table 1 ijms-25-09447-t001:** Purinergic signaling in hepatic stellate cells (HSCs).

HSCs		
Ligand, Receptor or Enzyme	Effect	Reference
Not determined	IP_3_ production and cytosolic Ca^2+^ mobilization and induction of cell contraction	[110]
P2Y2 and P2Y4 in HSCs and P2Y1 and P2Y6 in MFB	Cytosolic Ca^2+^ mobilization. Stimulation of activated HSCs with UDP increases the expression level of procollagen-1 by threefold.	[111]
P2X7	Increases expression level by acetaldehyde. Receptor stimulation with BzATP in HSCs induces upregulation of cell proliferation, inflammation and activation markers.	[112]
P2X4	The subunit was overexpressed in hepatotoxic-induced fibrotic liver and in the cell line HSC-T16 treated with acetaldehyde. Blocking of the receptor with 5-BDBD prevents activation of HSC-T16.	[89]
P2Y14	High expression in HSCs and their ligands, UDP-glucose and UDP-galactose, are released by dying hepatocytes. P2Y14 activation mediates HSC activation.	[113]
NTPDase2	It is expressed through the HSC activation process.	[111]
NTPDase1 (CD39) and NTPDase2	Expressed by HSCs cultured for 1 (quiescent), 4, and 7 (activated) days. In livers with fibrosis induced by administration of CCl_4_ for 6 weeks, NTPDase2 co-localizes with activated HSCs.	[111]
NTPDase1 (CD39)	Pharmacological or genetic inhibition of its expression in the T6 HSC line abolishes acetaldehyde-dependent proliferation and expression of fibrosis markers.	[114]
ENPP2-3	The enzymes were detected in the mouse GRX HSC line. Its activity is higher in the quiescent phenotype than in the activated one, and the expression level of NPP2 increased while NPP3 expression decreased.	[115]
CD73	CD73 KO mice are resistant to experimental fibrosis.	[116]
CD73	Its expression level increases throughout the HSC activation process; this regulation is mediated by the TGF-b pathway.	[117]
CD73	Plays a role in HSC activation, including when the process is mediated by alcohol-dependent damage models.	[118,119,120,121]
A2a	In the human HSC line LX2, A2a receptor activation inhibited the PDGF-induced chemotactic response by modulating a pathway that increased cAMP and upregulated the production of TGF-β and collagen I RNA.	[122]
A2a	Induces collagen type I and type III overexpression, mediated by a pathway that involves protein kinase A, src kinase, and mitogen-activated protein kinases ERK and p38 in the LX2 HSC line.	[123]
A2a	Interferon-g regulates A2a receptor function via the transcriptional regulation of AC by STAT-1 in LX2 cells.	[123,124]
A2a	Mediates loss of actin stress fibers throughout a pathway that inhibits cAMP, PKA, and Rho and suppresses endothelin-1 and lysophosphatidic acid-induced cell contraction in the LX2 HSC line.	[125]
A2a	Increased cell proliferation and inhibited senescence entry in the LX2 HSC line.	[126]
A2a	In the HSC-T6 line, the antagonists caffeine and ZM241385 inhibit the acetaldehyde-dependent expression of fibrotic markers by a cAMP-dependent pathway.	[126,127]
A2a	Caffeine inhibits the activation of primary mouse HSCs by Akt1 signaling.	[128]
Not determined	Adenosine aspartate prevents the activation of primary rat HSCs and inhibits the proliferative effect of platelet-derived growth factor.	[128,129]

**Table 2 ijms-25-09447-t002:** Purinergic signaling in Kupffer cells (KCs).

Kupffer Cells		
Ligand, Receptor or Enzyme	Effect	Reference
P2X4 and P2X6	Both subunits were detected in ED1+ KC by immunofluorescence. Intraperitoneal injection of LPS upregulated P2X6.	[149]
P2X7	Induces NLRP3 inflammasome assembly and, consequently, IL-1β release in primary KC.	[76]
P2X7	The environmental contaminant imidacloprid, a hepatotoxic compound, induces P2X7-dependent pyroptosis in primary KCs.	[151]
P2X7	In the KUP5 mouse line primed with LPS, stimulation with 3 mM of ATP induced channel activation and triggered the release of inflammatory messengers, such as prostaglandin E2, IL-1β, and high mobility group box-1 protein (HMGB1).	[152]
P2X7	The exposure of KUP5 cells to silica nanoparticles (30 nm in diameter) induced ATP release, activating the P2X7 receptor that induces IL-1ẞ production.	[153]
P2Y6	The receptor is overexpressed in the KCs of mice with alcohol-induced steatohepatitis. In the liver of these animals, the administration of UDP elevated the damage marker levels and the release of the proinflammatory cytokines.	[154]
ADO	ADO administration in ischemic preconditioning, a transitory occlusion of the portal triad and reperfusion prior to ischemia, protects liver graft viability by modulating KC-dependent inflammation.	[155,156]
A2	Alcohol feeding in mice induced A2b overexpression in KCs. This was reproduced in vitro in the RAW264.7 cell line treated with ethanol. In this model, the A2b receptor KO induces the expression of proinflammatory cytokines while overexpression of this receptor weakens the inflammatory response.	[157]
A2b	Alcohol feeding in mice induced A2b overexpression in KCs. This was reproduced in vitro in the RAW264.7 line treated with ethanol. In this last model, A2b KO boosts proinflammatory cytokine expression while overexpression weakens inflammatory response.	[158]
CD39	Myeloid cell-specific deletion of this enzyme increased mouse sensibility to pharmacologically induced biliary fibrosis, together with an increment in the expression levels of the transcripts of the inflammation markers.	[158,159]
CD39	CD39^−/−^ mice show higher inflammation levels and liver damage in response to sepsis, probably due to macrophage activity by deficient extracellular ATP scavenging.	[93]

**Table 3 ijms-25-09447-t003:** Purinergic signaling in liver sinusoidal endothelial cells (LSECs).

Liver Sinusoidal Endothelial Cells		
Ligand, Receptor or Enzyme	Effect	Reference
ATP and ADP	Stimulates the release of prostacyclin (PGI2) and nitric oxide (NO), inhibitors of platelet aggregation, through a mechanism involving Ca^2+^ mobilization.	[166,167]
P2Y and/or ADO receptors	Prostaglandin E2 (PGE2) is the main prostanoid secreted by primary cultures of LSECs in response to extracellular nucleotides.	[168]
CD39	Partial hepatectomy induces a specific increment in the expression level of CD39 in LSECs. In CD39^−/−^ mice, hepatic regeneration was impaired. Throughout the regenerative process, angiogenesis decreased while hepatic damage increased.	[168,169]
CD39	Co-culture of mouse LSECs expressing CD39 boosts the proliferation of tumor cells and limits extracellular ATP-dependent cell death induction.	[170]
A2a	The A2a agonist CGS21680 protected LSECs from the damage induced by ischemia reperfusion.	[170,171]

**Table 4 ijms-25-09447-t004:** Purinergic signaling in cholangiocytes.

Cholangiocytes		
Ligand, Receptor, or Enzyme	Effect	Reference
P2Y2	In cholangiocytes, UTP stimulates secretion.	[205,206]
P2Y12	The expression of the receptor in the apical membrane of cholangiocytes appears to be restricted to the cilium, a mechano-, osmo-, and chemo-sensory organelle that senses and transmits different stimuli from the bile into the cellular interior, modifying cell functions.	[208,209]
P2X: 2, 3, 4, and 6	In cultured rat cholangiocytes, they modulated biliary secretion.	[210]
P2X4	In Mz-ChA-1 cells, BzATP elicited robust Cl- secretory responses.	[210]
A2a	In polarized rat cholangiocytes, it modulates the activity of apical CNT3.	[212]
P2Y6, P2Y13, and P2X7	In cholangiocarcinoma cells, they induce proliferation and motility.	[213,214]
P2Y2 and P2X3	In MMNK-1, an immortalized cholangiocarcinoma cell line, this receptor conferred resistance to cell proliferation and cell motility inhibition.	[213,214]

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
