# Peer review of "Purinergic Signaling in Non-Parenchymal Liver Cells"

_ijms, 2024, doi:10.3390/ijms25179447_

Round 1

Reviewer 1 Report

Comments and Suggestions for Authors

Interesting theoretical topic and nice submitted review but lacking clinical relevance.

Major points:

1. Theorical background is perfectly presented.

2. Please add a section of therapy proposals for each liver liver disease, for which purins may be helpful. Consider DILI, HILI, ALD, NALD, Wilson disease, hemochromatosis, HBV, HCV, and many others. 

3. Focus on metabolic pathways and non-parenchymal as well as on  parenchymal cells as targets.

4. I am missing search terms.

5. Clarify at beginning of the main text the aim of the review article.

6. I am missing a short introduction with essentials  and aim. 

7. Minor. At one place, you say "As mentioned". Expression is less helpful, replace by what you really mean. 

Comments on the Quality of English Language

Some English improvement is needed. 

Author Response

Thank you very much for the careful and accurate review of our manuscript. Indeed, it has helped us to improve the perspective and quality of the document.

Reviewer 1

Comments and Suggestions for Authors

Interesting theoretical topic and nice submitted review but lacking clinical relevance.

Major points:

  1. Theorical background is perfectly presented.

Answer: Thanks.

  1. Please add a section of therapy proposals for each liver liver disease, for which purins may be helpful. Consider DILI, HILI, ALD, NALD, Wilson disease, hemochromatosis, HBV, HCV, and many others. 

Answer: a new section named “Therapeutic targeting of purinergic signaling in non-parenchymal liver cells” was added to the manuscript. It must be noted that the field of purinergic signaling in liver pathologies is sparse. For example, PubMed retrieved no reports in the search of “liver purinergic signaling” with “HILI”, “hemochromatosis” and “Wilson disease”; in exploring with the term “Drug induced liver injury (DILI) from a total of 47,993 entries, only 18 (0.037%) contained purinergic elements. However, we mentioned in the manuscripts some examples of therapeutic approaches involving the hepato-protector action of an adenosine derivative.  

  1. Focus on metabolic pathways and non-parenchymal as well as on  parenchymal cells as targets.

Answer: We have mentioned in the text some examples in which non-parenchymal hepatic cells were targets of therapeutic approaches but, in general, the information is quite scarce. This situation can be interpreted as to the field of purinergic signaling in physiopathology of non-parenchymal cells is incipient at present but promising for the near future.

  1. I am missing search terms.

Answer: The manuscript already has a list of key words.

  1. Clarify at beginning of the main text the aim of the review article.

Answer: A new section named “Introduction” and exposing the aim of the article was added.

  1. I am missing a short introduction with essentials  and aim. 

Answer: A new section named “Introduction” and exposing the aim of the article was added.

  1. Minor. At one place, you say "As mentioned". Expression is less helpful, replace by what you really mean. 

Answer: Done

Reviewer 2 Report

Comments and Suggestions for Authors

Authors summarized the purinergic signaling in non-parenchymal liver cells. The manuscript is a comprehensive and detailed review. I hold a positive attitude toward its acceptance. There are only few concerns.

Whether the key words are too many?

Line 164, please define “GLOBOCAN”.

Line 257, two spaces between “receptor” and “activation”.

Authors can consider to supply the descriptions on the endogenous and exogenous purine. Of course, it was no mandatory.

Compared with liver, kidney may have a more closed association with purine. What is the relationship between liver and kidney in purinergic signaling?

Author Response

Thank you very much for the careful and accurate review of our manuscript. Indeed, it has helped us to improve the perspective and quality of the document.

Reviewer 2

Comments and Suggestions for Authors

Authors summarized the purinergic signaling in non-parenchymal liver cells. The manuscript is a comprehensive and detailed review. I hold a positive attitude toward its acceptance. There are only few concerns.

Whether the key words are too many?

Answer: The number of key words was shortened

Line 164, please define “GLOBOCAN”.

Answer: Done

Line 257, two spaces between “receptor” and “activation”.

Answer: The mistake was corrected

Authors can consider to supply the descriptions on the endogenous and exogenous purine. Of course, it was no mandatory.

Answer: We consider that this description is not essential for the manuscript.

Compared with liver, kidney may have a more closed association with purine. What is the relationship between liver and kidney in purinergic signaling?

Answer: The question was addressed in the new section “Therapeutic targeting of purinergic signaling in non-parenchymal liver cells”, lines 848-845.

Round 2

Reviewer 1 Report

Comments and Suggestions for Authors

Thank you for good revision.

Comments on the Quality of English Language

Minor needed